# Side Bioimpedance Analysis in Menopausal Post-Oncological Breast Cancer

**DOI:** 10.3390/ijerph191811329

**Published:** 2022-09-09

**Authors:** Giuseppe Bifolco, Antonio Pinazzi, Vittorio Bini, Laura Stefani

**Affiliations:** 1Sports Medicine Center Clinical and Experimental Medicine Department, University of Florence, 50121 Firenze, Italy; 2Department of Medicine, University of Perugia, 06123 Perugia, Italy

**Keywords:** breast cancer, body composition, mastectomy

## Abstract

Background: Post-oncological BC (breast cancer) has an increased cardiovascular risk due to the variation of some anthropometric parameters. This study investigates the differences between a quadrantectomy and a mastectomy on the body composition over time in presence of a breast prothesis. Methods: A group of BC patients (n = 41 aged 56.6 ± 9.5 years; 15 mastectomy patients; and 26 quadrantectomy patients) were compared to controls (C) (n = 22 aged 46.5 ± 13.44 years). Through bioimpedance analysis (Akern-BIA 101), the body mass index (BMI), total body water (TBW), extracellular water (ECW), body cell mass (BCM), fat mass (FM), free fat mass (FFM), and angle phase (PA) were compared within each group and between different groups using the Student’s Test T. Results: The BC group showed lower values of FFM and TBW compared to C. The FFM was significantly (*p* = 0.04) lower in those with quadrantectomy. The right hemisome of the quadrantectomy has increased values of FFM, BCM (*p* = 0.04) and TBW compared to the counter-lateral hemisome, and FM values (*p* = 0.0008) lower than the counter-lateral. The hemisome with intervention has increased values of FM and ECW compared to the counter lateral, as well the FFM, BCM, TBW, and PA. Conclusions: The results support the hypothesis that non-conservative surgical treatment (mastectomy) is associated with a better BIA profile without any substantial impact of breast implants in the body composition analysis. The awareness of a severe diseases could play a role to ameliorate lifestyle; however, further studies will be necessary to support this theory.

## 1. Introduction

Breast cancer is the most frequent type of cancer in females [1]. Many subjects are submitted to surgical treatment in the form of a quadrantectomy or a mastectomy [2] in addition to chemotherapy (CT) with potential myocardial damage due to the cardiotoxicity. A progressive increase of body weight during chemotherapy has often been described [3,4]. The presence of fat mass can also support a potential recurrence of disease in them due to the presence of fat where the aromatase enzyme system can increase the estrogen level [5,6].

In order to maintain a normal heart function, to reduce the cardiovascular risk, and to preserve the high level of quality of life, regular physical activity is often prescribed, particularly at moderate intensity.

Sedentarism plays an important role in the lifestyle reconditioning, inducing comorbidities [7] especially in women submitted to a surgical treatment. This aspect could be enhanced, particularly for the subjects with prolonged exposition to a medical treatment in case of prosthesis implantation. Comorbidities can be found in the post-oncological time, especially if the cancer is coincident with the menopausal period.

This study was designed to investigate the first-line influence of the different kinds of surgical treatment, such as quadrantectomy and mastectomy, on the parameters of body composition over time. Its second aim was to evaluate the impact of a breast prosthesis implantation on the eventual alteration of the altered body composition parameters.

## 2. Materials and Methods

The population of this study included a case group (n = 41) and a control group (n = 22). An informal written consent, approved by the local ethical committee, was obtained, as is regularly performed for sports medicine and lifestyle evaluation. Considering the data are available in the dataset of the University Center, for the constant clinical check-up, no trial registral number was performed. The women in the case group (aged 56.6 ± 9.5 years) have a BMI of 26.1 ± 4.3 kg/m^2^ and are in menopause time; while the women of the control group were 46.5 ± 13.4 years old with a BMI of 24.88 ± 5.7 kg/m^2^, and they were perimenopausal–menopausal woman. Only three cases were in the fertile status. The case group was recruited by the Sport Medicine Center, because they followed a lifestyle reconditioning program after breast cancer recovery, while the control group were women volunteers without a breast cancer diagnosis. They underwent a body composition [8] study by bioimpedance analysis [9,10] following the guidelines of the standard tetrapolar technique using bioelectrical impedance analyzer (BIA 101 BIVA Pro, Akern, Florence, Italy). BIA analysis was performed in the Sport Medicine Center by the nutrition specialist. The data were obtained in the morning, after 10 min of rest condition. The data collection was during the period from February 2021 to July 2021. Regarding the control group, particularly in the few cases in the fertile status and in order to avoid the eventual modifications of the water distribution, due to the hormone’s level, the data were obtained, excluding the premenstrual phase or the menstruation. The women lay on the bed for about ten minutes in order to favor a homogeneous distribution of body fluids and without metal objects worn to avoid interference, in a quiet environment. A pair of electrodes at the hand level and another pair at the foot level were attached to them about 5 cm apart on the same side of the body. The same procedure was made on the counter–later side. 

In the first line, the body composition of post oncological patient was compared with the control group, and then the group of cases was further subdivided according to the type of surgery. The mastectomy patients (54.7 ± 8.8 years, BMI 23.7 ± 3.5 kg/m^2^) were compared with the control group, and then they were further subdivided according to the hemisome that underwent the surgery, after which they were compared with each other, then the right hemisome vs. the left hemisome, then they were compared with the control group, and finally with the contralateral hemisome that had not undergone any surgery. 

The same procedure was carried out for the patients who underwent a conservative treatment quadrantectomy, (57.8 ± 9.9 years; BMI 27.5 ± 4.1 kg/m^2^). They were first compared with the control group (Table 1), and then they were subdivided according to the hemisome that underwent the surgery; these were then compared with each other, with the contralateral (Tables 3 and 4), and finally with the control group. In addition, the brachial circumference in cm was also considered and compared, measured during the anthropometric assessment using a tape measure. A comparison of the two BC groups (conservative and not conservative surgical treatment) was made (Table 2).

### Body Composition Analysis—BIA Method and Procedure

Body impedance is generated in lean tissues as an opposition to the flow of an injected alternate current. Bioelectrical impedance was measured with phase-sensitive impedance plethysmography (BIA101 Sport Edition, Akern, Florence, Italy). The device emits an alternating sinusoidal electric current of 800 mA at a single operating frequency of 50 kHz, and standard whole-body tetra polar measurements were performed according to the manufacturer’s guidelines [11]. Resistance (Rz, Ω) is the opposition to the flow of an injected alternating current; Reactance (Xc, Ω) is the dielectric or capacitive component of cell membranes and organelles; Phase Angle (PA, in degree) is defined as the ratio between Rz and Xc or between intra- and extracellular volumes. From the values of Rz and Xc, through regression equations, the following body compartments are estimated: Fat-Free Mass (FFM), Extra Cellular Mass (ECM), Total Body Water (TBW), Extra Cellular Water (ECW), and Intra Cellular Water (ICW). A dedicated software provides the final data of the nutritional and hydration condition of the patient. Regarding the procedure followed, the electrodes were positioned in the middle of the dorsal surfaces of the hands and feet, proximal to the metacarpal–phalangeal and metatarsal–phalangeal joints, respectively, and also medially between the distal prominences of the radius and the ulna and between the medial and lateral malleoli at the ankle. Specifically, the proximal edge of one detector electrode was in line with the proximal edge of the ulnar tubercle at the wrist, and the proximal edge of the other detecting electrode was in line with the medial malleolus of the ankle. The current introducing electrodes are placed at a minimum distance of the diameter of the wrist or ankle beyond the paired detector electrode. The upper limbs were apart from the trunk (30°). The lower limbs were also apart (45°). In obese subjects, it was necessary to put an insulating cloth between the armpits and between the thighs. The subject did not move, the skin had not sweated and was cleansed with ethyl or isopropyl alcohol. The environment was ventilated or had low relative humidity. The room temperature was the typical medical office temperature, between 24–27 °C. Subjects had to have been fasting for at least 2 h, without having consumed alcohol. They were not to have taken diuretics and were not to be in a febrile state [12].

## 3. Statistical Analysis

The distributions of variables were first checked using the Shapiro–Wilk test, and, as they resulted in normal distributions, data were expressed as mean ± standard deviation (SD). The Student’s t tests for independent and paired data were used to analyze differences between groups and within groups, respectively.

All statistical analyses were performed using IBM-SPSS^®^ version 26.0 (IBM Corp., Armonk, NY, USA, 2019). In all analyses, a two-sided *p*-value ≤ 0.05 was considered significant.

## 4. Results

The comparison of the body composition of the women with quadrantectomy vs. the control group shows that the case group had lower FFM values than the controls (*p* = 0.04) and higher FM (*p* = 0.04) and brachial circumference values in cm for both limbs (*p* = 0.01) (Table 1).

Comparing women who had either a mastectomy and a quadrantectomy, increased brachial circumference values for both limbs in the latter were even compared to women with mastectomy either in the right side (29.5 ± 3.6 cm vs. 32.5 ± 2.9 cm; *p* = 0.05) as in the left side (*p* = 0.004) (Table 2).

In addition, the women who had had a quadrantectomy were further divided according to the hemisome that was surgically involved, and, in the comparison between the bioimpedance parameters measured for the right and left hemisome, it emerged that the right hemisome that underwent the surgery presented increased values of FFM (*p* < 0.001), BCM (*p* = 0.04) and TBW (*p* = 0.002) compared to the contralateral hemisome and lower FM values (*p* < 0.001) compared to the contralateral; overall, it therefore presents a better bioimpedance profile than the contralateral (Table 3).

The left hemisome, surgically treated, presented increased values of FM (*p* = 0.004) and ECW (*p* < 0.001) compared to the contralateral and lower values of FFM (*p* = 0.004), BCM (*p* < 0.001), TBW (*p* < 0.001), and PhA (°) (*p* < 0.001) compared to the contralateral (Table 4).

Then, the breast implant patients, most of whom had a mastectomy, were compared with the control group. No statistically significant differences emerged. The hemisomes with a breast implant were further compared with the other hemisomes of the same patient, again without statistically significant differences.

## 5. Discussion

Body composition has been largely studied in the literature, especially regarding breast cancer. The post-oncological time is of particular interest as consequence of the potential association with comorbidities, especially if the patient is in the menopausal time. The clinical implication of this aspect is related to the augmented cardiovascular risk [13,14] The different impact in the presence of a diverse surgical approach in breast cancer women has not, however, been investigated deeply in the literature. In addition, the eventual influence of a prosthesis on body composition analysis has not been clarified. The relationship of side-by-side bioimpedance analysis for the body composition assessment among quadrantectomy vs. mastectomy, in particular, has not yet been studied. The major significant results show evidence that, in the case of a non-conservative treatment (mastectomy), the body composition profile is better than in the case of a conservative treatment, as in quadrantectomy. In addition, the data in this last group show the hemisome that has been surgically treated has better values vs. the contralateral one; this is particularly evident in the right hemisome. This profile, on the contrary does not result in the comparison between the left hemisome that has undergone the operation vs. the contralateral one. The reason for this particular evidence is that the original aspect of the present investigation cannot be completely clarified from the present data; however, some aspect of the specific awareness of the severity of the disease could justify these results. All the values found were not out of the range reported in the literature, and this supports the hypothesis to obtain an objective analysis of the body composition in the women with a breast prosthesis. 

The results, therefore, are not completely in agreement with those found in a previous study assessing the interference of the breast implant, due to it being recognized as adipose tissue [15]. In any case, the group investigated is restricted, and this aspect could be considered a limit of the study for a large conclusion in BC. In the mastectomy group, it emerged that the body composition is better when compared to quadrantectomy.

In the first line, quadrantectomy, although it is considered conservative and involves the removal of only one breast quadrant, is apparently associated with bioimpedance parameters at a higher cardiovascular risk. This represents the most original aspect that has emerged from this study and has not been investigated before in the literature.

This apparent discrepancy could be attributed to a major emotive involvement on behalf of the women who have undergone a mastectomy and who carry out sporting activities such as rowing, tennis, and dragon boating mainly with the right limb. Therefore, the lifestyle aspect of the women under study could be better investigated in future.

It could be hypothesized that a partial or total removal of an organ, such as the breast, in addition to affecting the psychological state, could cause a change in lifestyle and physical activity due to the presence of pain, lymphoedema, paresthesia, decreased muscle strength, and a reduced range of motion of the limb involved. It can alter their perception of their well-being [16] and their quality of life [17,18,19]. Studies support the hypothesis that pain is a frequent sequela of breast-conserving surgery and radiotherapy, and that such symptoms may cause postoperative psychosocial distress, thus limiting the adaptation of the patient and reducing the beneficial effect of such surgery on body image.

The investigation has some limits, due to the slight difference of the age range of the control group vs. the BC group, despite the fact that they are in the same range of critical period of transition from peri–menopause to menopause. Another limit is represented by the restricted number of subjects investigated, which limited the possibility to investigate the potential impact of some variables on the data obtained. Among them, the time to cancer onset and diagnosis could be relevant, especially for the elaboration of the awareness of the disease.

## 6. Conclusions

Mastectomy and quadrantectomy are surgical treatments due to the severity of the disease, and often the management and the recommendations of the lifestyle of the women involved are similar. From the results obtained, it emerges that the cardiovascular profile of a quadrantectomy needs more attention than the mastectomy as demonstrated by the worst bioimpedance profile in the latter group.

This may be attributable to greater adaptation to the new reality and to a better quality of life than those who have had conservative surgery. Other studies have already shown that having had breast reconstruction surgery significantly increased the frequency of exercise [20]. The results also suggest that, with breast conservation rather than mastectomy, lymphoedema has become a very serious and frequent clinical problem despite attempts to minimize lymph node removal. This can lead to a reduced quality of life in surviving patients, as well as increased ECW values, and appears to be associated with an increased brachial circumference. 

Regarding the awareness of the disease, it has not been investigated in any depth in this study. Despite this limit, from the different results found in the group of the women with a mastectomy vs. a quadrantectomy, it could be hypothesized that it plays a role in determining a different approach to the women’s new life conditions, and, therefore, it could contribute to producing a better profile of those affected of a severe disease. A dedicated questionnaire to focus on this aspect could represent a feature investigation. 

In conclusion, a first consideration is that a quadrantectomy, although considered conservative and involving the removal of only one breast quadrant, is apparently associated with bioimpedance parameters at higher cardiovascular risk. This represents the most original aspect that emerged from this study and had not been investigated before in the literature.

Another important reflection is to consider minimizing this complication associated with breast cancer treatment; in addition to medical and pharmacological intervention, it will be necessary to promote a surveillance program that includes bioimpedance assessment, the prescription of an adequate diet plan, and a correct exercise program on the basis of the previous surgical treatment. This multidisciplinary program will not be addressed to achieve a normal weight condition, which is important for reducing cardiovascular risk factors and breast cancer recurrence, while tailoring the kind of the program in terms of major or minor cardiovascular risk. 

## Figures and Tables

**Table 1 ijerph-19-11329-t001:** Body composition of patients with quadrantectomy vs. control group.

	Case Group: Quadrantectomy (n = 26)	Control Group (n = 22)	*p*-Value
BMI, kg/m^2^	27.5 ± 4.1	24.88 ± 5.7	0.06
Circumference right arm, cm	32.2 ± 3.1	29.5 ± 4.2	0.01
FFM, %	66.0 ± 6.6	70.6 ± 9.0	0.04
FM, %	34 ± 6.6	29.2 ± 9.0	0.04
Circumference left arm, cm	32.4 ± 3.3	29.3 ± 3.9	0.01

Legend: BMI: Body Mass Index; FFM: Fat Free Mass; FM: Fat Mass.

**Table 2 ijerph-19-11329-t002:** Body composition «of side» in patients with mastectomy vs. patients with quadrantectomy.

	Case Group: Mastectomy Left (n = 6)	Control Group: Quadrantectomy Left (n = 15)	*p*-Value
BMI, kg/m^2^	22.4 ± 2.8	26.7 ± 4.7	0.05
Circumference left arm, cm	27.3 ± 1.6	31.7 ± 3.0	0.004

Legend: BMI: Body Mass Index.

**Table 3 ijerph-19-11329-t003:** Body composition «of side» in patients with quadrantectomy vs. contralateral.

	Case Group: Right Hemisome with Quadrantectomy (n = 11)	Control Group: Contralateral (n = 11)	*p*-Value
FFM, %	66.7 ± 5.8	65.0 ± 5.7	0.0008
FM, %	33.2 ± 5.8	34.9 ± 5.7	0.0008
BCM, %	54.3 ± 4.2	53.4 ± 3.2	0.04
Total body water, %	48.9 ± 4.3	47.6 ± 4.1	0.002

Legend: FM: Fat Mass; BCM: Body Cell Mass; ECW: Extra Cellular Mass.

**Table 4 ijerph-19-11329-t004:** Body composition of «side» left in patients with quadrantectomy vs. contralateral.

	Case Group: Left Hemisome with Quadrantectomy (n = 15)	Control Group: Contralateral (n = 15)	*p*-Value
PhA, (°)	5.7 ± 0.4	6.0 ± 0.5	0.0006
FFM, %	66.7 ± 7.4	68.0 ± 6.7	0.004
FM, %	33.3 ± 7.4	31.9 ± 6.7	0.004
BCM, %	52.4 ± 2.7	53.9 ± 2.8	0.0002
Total body water, %	48.8 ± 5.4	49.8 ± 4.9	0.0009
ECW, %	46.9 ± 2.5	45.5 ± 2.5	0.0002

Legend: PhA: Phase Angle; FFM: Fat Free Mass; FM: Fat Mass; BCM: Body Cell Mass; ECW: Extra Cellular Mass.

## Data Availability

The data are collected in the database of the sports medicine center—University of Florence.

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
