# Peer review of "Side Bioimpedance Analysis in Menopausal Post-Oncological Breast Cancer"

_ijerph, 2022, doi:10.3390/ijerph191811329_

Round 1

Reviewer 1 Report

The manuscript “Side Bioimpedance analysis in breast cancer” reports comparing the results of side bioimpedance analysis of patients subjected to quadrantectomy and mastectomy in menopause and the control group of apparently healthy women of childbearing age. The authors found that, in general, the patients had higher free fat index and brachial circumference values than those of the control group. The parameters of patients depending on the hemisphere undergone surgery were also studied.

Despite the fact that the topic of the paper is extremely important, it cannot be recommended for publication without major revision.

First, it is not entirely correct to compare the results for menopausal patients with those of the control group in a fertile state. In addition, it is doubtful to compare data for younger patients after mastectomy and older patients after quadrantectomy. For example, an increase in the brachial circumference values may be caused by a worse physical condition due to more advanced age, but also due to indifference to one's physical form associated with the awareness of the disease. The authors should not be silent about these points and express their ideas as a difference in the physiological age and stage at which the woman was subjected to surgery can have a side effect in processing the results.

The method contains references to the description of bioimpedance analysis but it is necessary to provide at least a brief description of its mechanism. Research results are poorly visualized. It is desirable to provide an analysis scheme, i.e. the position of the woman and the exact location of the electrodes. It will also be more convenient for the perception of information to depict the division of women into groups depending on the area of ​​surgery and to compile data for the right and left hemispheres.

Statements related to the assessment of the impact of the implant are not confirmed by any analysis, so they arise unexpectedly and unreasonably. Authors must provide either some experimental evidence or exclude such claims.

Please double check misprints and grammar mistakes, there are a few of them.

Author Response

REVISORE 1

The manuscript “Side Bioimpedance analysis in breast cancer” reports comparing the results of side bioimpedance analysis of patients subjected to quadrantectomy and mastectomy in menopause and the control group of apparently healthy women of childbearing age. The authors found that, in general, the patients had higher free fat index and brachial circumference values than those of the control group. The parameters of patients depending on the hemisphere undergone surgery were also studied.

Despite the fact that the topic of the paper is extremely important, it cannot be recommended for publication without major revision.

 We want to thank the reviewers  for giving us the opportunity to improve the manuscript  and to ameliorate  the comprehension and the interest for readers  . All the suggestions have ben accepted and the modifications requested have been made.

First, it is not entirely correct to compare the results for menopausal patients with those of the control group in a fertile state.

We thank for this  observation that offers  us  the possibility to clarify  this point .  Despite the different  age of the groups  analyzed , however we respectfully underline that the  control group is  composed  of women aged aged 46.5 ± 13.44 years ( from 46 up to 59 yrs) – and therefore   in a range of the perimenopausal- menopausal  time ( only 3 cases were  in the  fertile staus) , not substantially different  from the BC group  aged since 56.6  up to 64  years. We believe that     this a group  of women   cannot  be considered  in a childbearing age . This aspect has been clarified in deep  in the  text   and the details  of  the data  collection have been explained .   Following your suggestions  we  have modified the  text  in the material session and  discuss  this  aspect as  a potential  limit of the  study . The title  could be  also modified following  your kind observation .   

In addition, it is doubtful to compare data for younger patients after mastectomy and older patients after quadrantectomy. For example, an increase in the brachial circumference values may be caused by a worse physical condition due to more advanced age, but also due to indifference to one's physical form associated with the awareness of the disease. The authors should not be silent about these points and express their ideas as a difference in the physiological age and stage at which the woman was subjected to surgery can have a side effect in processing the results.

Thank  you  for this  observation, however We do  not understand exactly what do you mean.   The text reports data of The mastectomy patients (54,7 ± 8,8 years, BMI 23,7 ± 3,5 Kg/m²) and …. quadrantectomy (57,8 ± 9,9 years; BMI 27,5 ± 4,1 Kg/m²). We believe they can be  considered  similar.  Regarding the potential role  of  the  awareness of the  disease  and also  of the  kind of the surgical treatment , we are in agreement  with you. This  is  the  crucial poin of the  study  and  the    aim  of  the results. In any  any case  , from  this pilot  study,  the   data support the mastectomy is associated to a better body  composition  if compared to quandrantectomy . This  aspect has been discussed in the conclusions. It  could be reasonable to hypothesize that  a more invasive surgical treatment induces a stronger  awareness  of the clinical status  and therefore a stronger  response in terms  of lifestyle habits .

The method contains references to the description of bioimpedance analysis but it is necessary to provide at least a brief description of its mechanism. Research results are poorly visualized. It is desirable to provide an analysis scheme, i.e. the position of the woman and the exact location of the electrodes.

Thank  you  for his suggestions , more details  of the  BIA analysis has been  insert .  An additional reference has been insert .

 It will also be more convenient for the perception of information to depict the division of women into groups depending on the area of ​​surgery and to compile data for the right and left hemispheres.

We respectfully  underline  that  the  design of the study has been made following  this  scheme . …. In first line, the body composition of post oncological patient was compared with the control group and then, the group of cases was further subdivided according to the type of surgery

Statements related to the assessment of the impact of the implant are not confirmed by any analysis, so they arise unexpectedly and unreasonably. Authors must provide either some experimental evidence or exclude such claims.

We  are  in agreement  with  you. This  is just  an  hypothesis . This statement has been  modified in the text .

Please double check misprints and grammar mistakes, there are a few of them.

 The English version has been corrected 

Reviewer 2 Report

This paper describes bioimpedance and anthropometric results from a small group of women who have been treated for breast cancer, and a control group. This topic is of interest to many in this field where good quality data are required.

The selection of patients and data collection have been done appropriately. Unfortunately, the authors have performed far too many t-tests, making the interpretation of the results very difficult. 

As a recommendation, the authors should rewrite the manuscript focussing on their "best" result, namely:

"...quadrantectomy , although considered conservative and involving the removal of only one breast quadrant, is apparently associated with bioimpedance parameters at higher cardiovascular risk. This represents the most original aspect emerged from this study and not investigated before in literature. "

In addition, the authors should consider including the per-patient data in a supplement.

Author Response

This paper describes bioimpedance and anthropometric results from a small group of women who have been treated for breast cancer, and a control group. This topic is of interest to many in this field where good quality data are required.

The selection of patients and data collection have been done appropriately. Unfortunately, the authors have performed far too many t-tests, making the interpretation of the results very difficult. 

As a recommendation, the authors should rewrite the manuscript focussing on their "best" result, namely:

"...quadrantectomy , although considered conservative and involving the removal of only one breast quadrant, is apparently associated with bioimpedance parameters at higher cardiovascular risk. This represents the most original aspect emerged from this study and not investigated before in literature. "

 Thank you  for  this  suggestion that offer us  the opportunity  to reduce  the  length  of the  paper . However this suggestions is  in contrast  with the other reviewers , For him the details  of  the   groups  analysed ,  subdivided  in additional groups  for comparing the differences of the surgical  treatment , in  not sufficient . We  want  to to highlight your suggestion including  this  strong evidence  in the  discussion

In addition, the authors should consider including the per-patient data in a supplement.

 The per- patients data  are available  from the  data collection  of  the ambulatory setting , where  the  study has been performed as declared  in the text and in the consent form . Let  me know  if you need  an anonymous excel format .In any case  we respectfully underline that it could be  too havy . 

Round 2

Reviewer 1 Report

The clarifications and corrections made by the authors are convincing enough to recommend the manuscript for publication.